# Classification of Epithelial Ovarian Carcinoma Whole-Slide Pathology Images Using Deep Transfer Learning

**Yiping Wang**[*1,2]                                       YIPINGWANG@UVIC.CA

**David Farnell**[*1,3]                                     DAVID.FARNELL@VCH.CA

**Hossein Farahani**[*1,2]                                  H.FARAHANI@UBC.CA

**Mitchell Nursey**[1,2]                                    MNURSEY@UVIC.CA

**Basile Tessier-Cloutier**[1,3]           BASILE.TESSIERCLOUTIER@MAIL.MCGILL.CA

**Steven J.M. Jones**[4]                                    SJONES@BCGSC.CA

**David G. Huntsman**[1,4]                              DHUNTSMA@BCCANCER.BC.CA

**C. Blake Gilks**[1,3]                                     BLAKE.GILKS@VCH.CA

**Ali Bashashati**[1,2]                                     ALI.BASHASHATI@UBC.CA

[1] *Department of Pathology and Laboratory Medicine, University of British Columbia, Canada*

[2] *School of Biomedical Engineering, University of British Columbia, Canada*

[3] *Vancouver General Hospital, Canada*

[4] *BC Cancer Research Center, Canada*

## Abstract

Ovarian cancer is the most lethal cancer of the female reproductive organs. There are 5 major histological subtypes of epithelial ovarian cancer, each with distinct morphological, genetic, and clinical features. Currently, these histotypes are determined by a pathologist's microscopic examination of tumor whole-slide images (WSI). This process has been hampered by poor inter-observer agreement (Cohen's kappa 0.54-0.67). We utilized a *two*-stage deep transfer learning algorithm based on convolutional neural networks (CNN) and progressive resizing for automatic classification of epithelial ovarian carcinoma WSIs. The proposed algorithm achieved a mean accuracy of 87.54% and Cohen's kappa of 0.8106 in the slide-level classification of 305 WSIs; performing better than a standard CNN and pathologists without gynecology-specific training.

**Keywords:** Transfer learning, Ovarian cancer, Digital pathology

## 1. Introduction

Ovarian cancer ranks fifth in cancer deaths among women (Siegel et al., 2016), accounting for more deaths than any other cancer of the female reproductive system in North America. There are 5 major histological subtypes of epithelial ovarian cancer: high-grade serous ovarian carcinoma (HGSOC), clear cell ovarian carcinoma (CCOC), endometrioid (ENOC), low-grade serous (LGSOC) and mucinous carcinoma (MUC). These five major histotypes have distinct morphological, molecular, genetic, and clinical features (Köbel et al., 2008). Accuracy of pathologists in ovarian cancer histotype classification based on histo-morphological features of the tissue is hampered by poor diagnostic reproducibility and interobserver disagreement (Gilks et al., 2013; Clarke and Gilks, 2010; Han et al., 2013).

---

[*] Contributed equally

Without gynecologic pathology-specific training, which reflects most current pathology practices, the interobserver agreement is moderate, with Cohen's kappa varying between 0.54 and 0.67 (Köbel et al., 2013; Patel et al., 2012). Deep learning applied to image analysis, enabled by the digitization of pathology materials, provides an opportunity to revisit the rich information present in histopathology images, and improve ovarian cancer diagnosis.

## 2. Dataset

We collected and annotated a dataset of 305 whole-slide images (WSI) composed of 157 HGSOC, 53 CCOC, 55 ENOC, 29 LGSOC, and 11 MUC slides from the Vancouver General Hospital. The aforementioned WSIs originated from 159 ovarian cancer patients (76 HGSOC, 32 CCOC, 28 ENOC, 14 LGSOC, 9 MUC), for which the histological subtypes were determined by molecular assays and reviewed by several pathologists. Representative areas of tumor in each WSI were annotated by a board-certified pathologist. Due to the prohibitively large size of the WSIs, we tiled these annotated regions to patches of size $1024 \times 1024$ pixels at $40\times$ magnification (equivalent to $0.25\mu m/pixel$) leading to an average of 530 patches per slide. We then down-sampled the patches to $512 \times 512$ and $256 \times 256$ using Lanczos filter (Turkowski, 1990). For evaluation, we utilized a 3-fold cross-validation scheme, in which we randomly divided the dataset into three patient groups. Two of the three groups were used as the training set, with the remaining group divided equally by patient and alternatively swapped for validation and test sets.

## 3. Method and Results

Most classification methods for WSI employ a patch-based approach in which the classification is performed on every single patch (e.g., $256 \times 256$ pixels) and an aggregation method (e.g., majority vote) is used to predict the label of the WSI from all the patch-level labels. However, a small patch with a limited field-of-view (FOV) might lose the context of the morphological patterns which are required to accurately classify a WSI. To extend the FOV while considering the computational limits, tiled patches are usually down-sampled from high-resolution to low-resolution. The trade-off between the size of FOV and the resolution makes it difficult to balance the context and details in images. Inspired by ProGAN (Karras et al., 2018) and www.fast.ai, this paper proposes a *two*-stage deep transfer learning patch-level classification method using progressive resizing for pathology image analysis (Figure 1), which is not only able to extract the features from large FOV low-resolution patches, but is also able to further extract the small details from high-resolution patches. Finally, we aggregated the patch-level classification to slide-level classification using ensemble learning.

In *Stage 1*, we fed the low-resolution patches (i.e., $256 \times 256$ pixels) into a pre-trained VGG19 (Simonyan and Zisserman, 2015) network and replaced the last 1000-class fully-connected (FC) layer with a 5-class FC layer. A softmax layer was then applied at the end to obtain the categorical distribution corresponding to the five subtypes of ovarian cancer. Although a network trained on low-resolution images will potentially miss subtle features available from high-resolution images, it will likely pick up high-level contextual patterns that are critical for diagnosis and only available in larger FOV.

In *Stage 2*, we removed the first convolutional block of VGG19 and added two randomly initialized convolutional blocks at top of the trained network from *Stage 1* and fed the high-resolution patches (i.e., $512 \times 512$ pixels) into the resulting network. *Stage 2* takes advantage of the contextual features learned from low-resolution images, plus other essential small details from high-resolution images to further increase performance.

To aggregate the patch-level classification to slide-level classification, we created a matrix $\mathbb{Z}^{N \times C}$, where $N$ is the number of WSIs and $C$ is the number of histotypes. We assigned each patch the label corresponding to the results of the *Stage 2* classifier. Finally, to predict the WSI-level labels, we employed a random forest (RF) classifier using the following strategy: (a) for each slide, we extracted a 5-dimensional feature vector with each element representing the number of patches classified as one of the five histotypes of ovarian cancer followed by Z-score normalization, and (b) in a cross-validation strategy (using the exact similar cross-validation sets used for patch-level classification), we trained the RF classifier on various subsets of the data and tested its performance on the held-out set.

To demonstrate the utility of our proposed approach of transfer learning, we compared the performance against two approaches as baselines: (1) a conventional CNN in which we trained a VGG19 network (initialized by random weights) using $512 \times 512$ patches, and (2) Stage 2 network initialized with random weights. Table 1 shows the per-class and overall performance of the patch-level and slide-level classifiers.

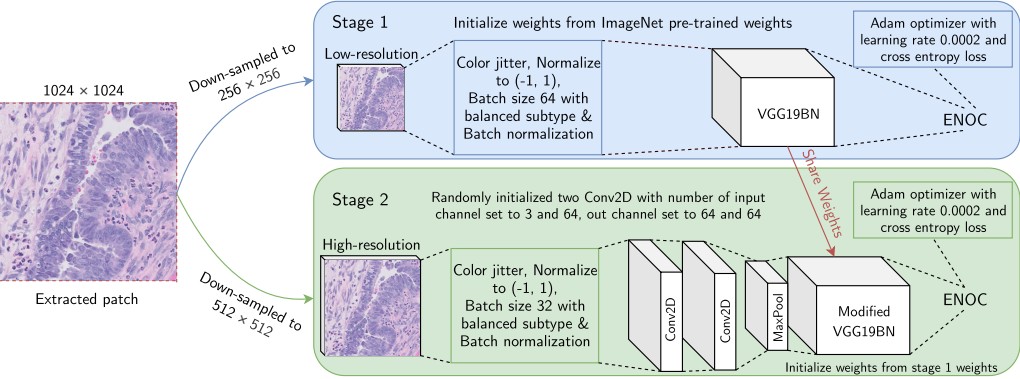

Figure 1: Schematic of the proposed *two*-stage deep transfer learning workflow

## 4. Conclusion

We proposed a transfer and ensemble learning approach for the automatic histological classification of epithelial ovarian cancer WSIs. The proposed algorithm performed better than either standard CNNs (across various performance measures) or pathologists without gynecologic pathology-specific training. These results suggest a promising future direction to validate the findings on a larger cohort of patients as well as explore other deep learning architectures that incorporate features from different magnifications and patch sizes. Furthermore, since most of the performance gain is observed in the slide-level classification results, we will compare the patch-level classification results (e.g., 5-dimensional feature vectors extracted from patch-level results) of various runs to get a better sense of the changes in

| Classifier | Mean Per-Class Accuracy | | | | | Overall | | | |
|---|---|---|---|---|---|---|---|---|---|
| | HGSOC | CCOC | ENOC | LGSOC | MUC | Accuracy | Kappa | AUC | F1 Score |
| Baseline 1 Patch-Level | 67.15% | **90.44%** | 62.79% | **71.00%** | 55.96% | 69.83% | 0.5992 | **0.9120** | 0.6850 |
| Baseline 2 Patch-Level | 62.76% | 81.10% | 59.51% | 52.34% | 53.31% | 62.63% | 0.5024 | 0.8410 | 0.6184 |
| Stage 1 Patch-Level | **74.94%** | 84.04% | **67.89%** | 61.81% | 59.98% | **71.75%** | **0.6187** | 0.9035 | 0.6984 |
| Stage 2 Patch-Level | 71.67% | 88.77% | 62.68% | 68.41% | **60.71%** | 71.60% | 0.6179 | 0.8890 | **0.7047** |
| Baseline 1 Slide-Level | 80.25% | 83.02% | 54.55% | 65.52% | 54.55% | 73.77% | 0.5993 | 0.9391 | 0.6855 |
| Baseline 2 Slide-Level | 80.13% | 75.47% | 34.55% | 68.97% | 54.55% | 69.08% | 0.5224 | 0.8481 | 0.6479 |
| Stage 1 Slide-Level | 85.99% | 79.25% | 61.82% | 79.31% | 54.55% | 78.69% | 0.6730 | 0.9375 | 0.7414 |
| Stage 2 Slide-Level | **90.45%** | **86.79%** | **74.55%** | **100.0%** | **81.82%** | **87.54%** | **0.8106** | **0.9641** | **0.8718** |

Table 1: Patch- and slide-level performance measured by various metrics in 3-fold cross-validation. Baseline 1 represents the results for a conventional VGG19 network trained on $512 \times 512$ patches. Baseline 2 represents the results for the randomly initialized Stage 2 network trained on $512 \times 512$ patches. Multi-class AUC (area under the curve) was calculated based on the approach proposed in (Hand and Till, 2001)

patch-level classification that contributed to better performance in slide-level classification. Code available at https://github.com/AIMLab-UBC/MIDL2020.

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
