# OpenReview forum: "Classification of Epithelial Ovarian Carcinoma Whole-Slide Pathology Images Using Deep Transfer Learning"
_MIDL.io/2020/Conference — MIDL 2020_

### Official Review · AnonReviewer2 · 2020-02-24
**Good paper, requires some further clarifications**

**Rating:** 4
**Confidence:** 5

**Review:**

This paper is well written, the methods are appropriate, sufficient detail is provided to enable replication, and the results are presented clearly. This paper could be improved by expanding on the motivation for this work, including information about how ground truth labels were obtained, and contextualizing this study with previous literature.

Detailed comments:

No motivation is given for how better classification of these subtypes would improve patient outcome.

The rationale for the two stage approach seems to be to incorporate features learned on the low-resolution space in learning the high-resolution space, and for using information from multiple scales to classify the patient. This approach is appropriate for the domain (pathological analysis) and is explained clearly and completely here. However I wonder why other approaches that use information from multiple domains, such as U-Net, were not considered.

Did all the images come from the same hospital? At what micron-per-pixel resolution were images digitized?

An important missing piece of this manuscript is a description of how the image class labels were determined. If this was done by an objective method, such as molecular analysis, there is no problem and a statement explaining the label origin can be added. However, if the class labels were based on morphological analysis, there is the problem that the labels are not true ground truth. If the ground truth labels are obtainable, such as through a molecular test, motivation needs to be provided for why this study is necessary.

With unbalanced classes, measures such as sensitivity and specificity or true positive rate and true negative rate should be reported instead of accuracy.

There are multiple methods for calculating AUC in a multi-class problem. The authors should state the method they used.

The authors state that their method outperformed the baseline method. This is only true in the slide-level case. I agree with the authors implicit assumption that patch-level metrics are less clinically important than slide-level metrics, but this should be made explicit and justified.

The conclusions of this work, that the two-stage method is better than conventional approaches, would be strengthened by referencing past publications that reported worse performance in this task.

---

### Official Review · AnonReviewer4 · 2020-03-03
**multi-resolution approach to classifying epithelial ovarian carcinoma**

**Rating:** 4
**Confidence:** 5

**Review:**

In the paper, the authors presented clear goals and well-designed experiments. Both, experiments and methods are well described and easy to follow.  The paper proposed a multi-resolution approach that is using two DL models and shared weights. The proposed approach is interesting.  The authors achieved decent results with the Kappa score equal to 0.8.
Adding the following information could be useful for readers:
-	How many areas were annotated per slide?
-	How the random forest classifier was trained (on which dataset)?
-	Missing references to the Lanczos filter.

---

### Official Review · AnonReviewer1 · 2020-03-10
**Deep transfer learning for histotyping of ovarian cancer using whole-slide images**

**Rating:** 4
**Confidence:** 4

**Review:**

- Reasonably well motivated problem, even if the clinical relevance isn't mentioned.
- Well described approach for 2-stage deep learning that trains on low and high detail images of the WSI.
- Comparison shows improved performance over standard patch based approach, though that too appears to have been run in a 2-stage setting.

---

### Official Review · AnonReviewer3 · 2020-03-14
**An effective transfer learning strategy**

**Rating:** 3
**Confidence:** 5

**Review:**

The manuscript proposes a 2-stage transfer learning strategy for the classification of epithelial ovarian cancer subtypes. The approach takes 1024×1024 patches from the whole slide image and downsample the patches to 256×256 to train a Stage-1 network in VGG-19 structure. The network is then embedded into a Stage-2 network with downsampled 512×512 as initial input. Results show that the proposed 2-stage strategy outperforms a baseline VGG-19 network and a Stage-1 network alone at whole slide level.
The manuscript is written clearly and the proposed approach shows its effectiveness.
Concerns that need to be addressed are:
1.	The Stage-2 network generates mixed results at patch level according to Table 1, while when integrated into whole slide level prediction the performance is improved significantly. Explanations are expected.
2.	It is worth evaluating the performance of training the Stage-2 network from scratch.
3.	It is unclear whether the patches fed into the baseline VGG network are downsampled or at original resolution.
4.	How will the model perform if 512×512 patches are used at original resolution?

---

### Meta-Review · Area_Chair1 · 2020-03-31
**MetaReview of Paper104 by AreaChair1**

**Rating:** 4

**Metareview:**

All reviewers agree that the paper is strong enough for acceptance. They highlight the interesting methodological developments, the well though-out experiments, and the easy-to-follow description. The results look promising as well.

**Paper Type:**

methodological development

---

### Decision · Program_Chairs · 2020-04-11

Accept